# Incidence, Impact and Treatment of Ongoing CMV Infection in Patients with Biliary Atresia in Four European Centres

**DOI:** 10.3390/jcm11040945

**Published:** 2022-02-11

**Authors:** Björn Fischler, Piotr Czubkowski, Antal Dezsofi, Ulrika Liliemark, Piotr Socha, Ronald J. Sokol, Jan F. Svensson, Mark Davenport

**Affiliations:** 1Astrid Lindgren Children’s Hospital, Karolinska University Hospital, CLINTEC Karolinska Institutet, 171 64 Stockholm, Sweden; ulrika.liliemark@karolinska.se; 2The Children’s Memorial Health Institute, 04 730 Warsaw, Poland; p.czubkowski@ipczd.pl (P.C.); p.socha@ipczd.pl (P.S.); 3First Department of Pediatrics, Semmelweis University, 1085 Budapest, Hungary; dezsofi.antal@med.semmelweis-univ.hu; 4Pediatric Liver Center, Department of Pediatrics, University of Colorado School of Medicine, Children’s Hospital Colorado, Aurora, CO 80045, USA; ronald.sokol@childrenscolorado.org; 5Astrid Lindgren Children’s Hospital, Karolinska University Hospital, Department of Women’s and Children’s Health, Karolinska Institutet, 171 64 Stockholm, Sweden; jan.f.svensson@karolinska.se; 6Kings College Hospital, Department of Paediatric Surgery, London SE5 9RS, UK; markdav2@ntlworld.com

**Keywords:** biliary atresia, cytomegalovirus, survival native liver, antiviral treatment

## Abstract

Cytomegalovirus (CMV) infection has been suggested to be of importance for the development and outcome of biliary atresia (BA). However, most data are only available from single centre studies. We retrospectively collected data on rates, outcomes, and treatments for ongoing CMV infection at the time of Kasai portoenterostomy (KPE) from four different tertiary centres in Europe. The rate of ongoing CMV infection varied between 10–32% in the four centres. CMV positive patients were significantly older and had higher levels of several liver biochemistries at the time of KPE (*p* < 0.05 for all comparisons). In the largest centre, CMV infection was more common in non-Caucasians, and CMV infected patients had poorer long-term survival with native liver than CMV negative patients (*p* = 0.0001). In contrast, survival with native liver in the subgroup of CMV infected patients who had received antiviral treatment was similar to the CMV negative group. We conclude that ongoing CMV infection at the time of KPE occurs in a significant proportion of BA patients and that these patients seem to differ from CMV negative patients regarding age and biochemistry at the time of KPE as well as long-term survival with native liver. The latter difference may be reduced by antiviral treatment, but randomized, controlled trials are needed before such treatment can be recommended routinely.

## 1. Introduction

Cytomegalovirus (CMV) is a double-stranded DNA virus of the family *herpesviridae*. Though infection is normally mild in children and adults it can be serious in the immunocompromised host. Congenital CMV infection, in contrast, can be a significant cause of microcephaly, neurodevelopmental delay and hearing loss and is said to be present in 1–2% of all pregnancies [1].

The role of viruses in initiating or causing biliary atresia (BA) has been debated for at least 30 years but without a definitive consensus. Of all the possible viruses proposed, CMV appears to have the strongest evidence [1,2,3]. In an early Swedish study from 1998, CMV-IgM was detected in serum from 38% of BA patients at the time of Kasai portoenterostomy (KPE), which was significantly higher than the 6% found in age-matched controls without any liver disease [1]. Other studies have since suggested the rate of BA patients with ongoing CMV infection to be anywhere between 10 and 74% with Asian centres tending to have a higher prevalence [2,3,4,5,6]. In the largest European study to date, Zani et al. showed that infants with “CMV-IgM +ve BA” were older at the time of KPE and had distinctively different histopathological features in the liver, including more pronounced inflammation than BA patients without CMV infection [3]. It was also reported that CMV IgM +ve patients had a worse prognosis that was improved following anti-viral treatment (AVT) [7].

Currently, there is a lack of multicentre studies on the frequency, consequences and possible importance of CMV infection in BA. The aim of the present study was therefore to collect data from four European tertiary centres on the rate of ongoing CMV infection at the time of KPE, identify the differences between CMV infected and uninfected BA patients at the time of KPE and their outcomes, and to examine the possible effects of AVT.

## 2. Materials and Methods

Clinical and biochemical data were retrospectively collected from locally held databases and patient charts at the following centres in Europe: First Department of Pediatrics, Semmelweis University, Budapest; Kings College Hospital, London; Astrid Lindgren Children’s Hospital, Karolinska University Hospital, Stockholm; and Children’s Memorial Health Institute, Warsaw (Table 1).

Ongoing CMV infection (CMV positivity) detected before or at the time of KPE was defined by any of the following: a positive test for serum CMV-IgM, urine CMV-DNA by PCR or CMV-DNA in serum/plasma/whole blood by PCR.

CMV-positive BA patients were compared to CMV-negative patients with regard to ethnicity, associated anomalies, age and biochemical lab values at KPE and outcomes. For CMV-positive patients, treatment and outcome of antiviral treatment were recorded. For the largest centre (London), 50 CMV-positive patients were compared to 100 contemporaneous CMV-negative control BA patients. Two control patients were matched in time to one index CMV-positive case, to correct for practice and surgery at the time.

Data were reported as median (range) unless otherwise indicated. Differences were tested using non-parametric statistical tests and a *p* value of <0.05 was accepted as statistically significant.

In Stockholm and Warsaw, the study was approved by the local ethics committees. In Budapest and London, data collection was regarded as an audit of outcome and, thus, formal ethical approval was not required. Informed consent was waived in Stockholm and Warsaw since it was not required according to the ethical permit. Informed consent was obtained in Budapest.

## 3. Results

All four centres reported data from the past 15 years with one (Warsaw) dating back to 1990 (Table 1); the reported prevalence of CMV testing in BA patients was high in all four centres (67–100%) and was universal in one centre (Stockholm). The rate of ongoing CMV infection at the time of KPE varied between an estimated 10% in London to 32% in Stockholm and was associated with the age at KPE (Table 2 and Table 3). In Budapest, Stockholm and Warsaw, 81 out of a total of 407 BA patients (19.9%) had signs of ongoing CMV infection at the time of KPE. In London, an estimated 10% of all BA were CMV positive. For the purpose of further analysis 50 patients with ongoing CMV and 99 CMV-negative controls were chosen. Thus, altogether 180 CMV-positive BA patients from four centres were analysed.

Age at KPE was significantly higher in CMV positive than in CMV negative BA patients in three of the centres (all *p* < 0.05), with borderline significance in the fourth (Table 3). Available liver biochemistries at the time of KPE were often significantly higher in the CMV-positive group compared to the CMV-negative group although this was variable. For instance, CMV-positive infants were significantly more jaundiced in London than at the other three centres. APRi (a surrogate marker of liver fibrosis) was significantly elevated in two of the three centres where it was measured in infants with CMV-positive BA (Table 2).

In only one of the centres (London) was there an ethnic difference with CMV positivity being significantly more common among non-Caucasian patients than among Caucasians.

The effect of CMV on clinical outcomes was more difficult to delineate as two centres did not report jaundice clearance rates. Clearance was significantly lower in CMV-positive cases (37% vs. 71%; *p* < 0.0001) in London, while not statistically different in Stockholm (25% vs. 44%; *p* = 0.26) (Table 4). Any differences in native-liver survival were also unclear in some centres. However, it should be noted that there was no reported native-liver survival in CMV-positive infants in Budapest. Figure 1 illustrates a significant difference in native-liver survival according to CMV status in the London cohort with the largest number of patients; however, this was not consistently reported in the other centres.

AVT (i.v. ganciclovir and/or oral valganciclovir) was widely used in CMV-positive patients in three centres (50–92% of CMV positive infants), but in only 25% of the London series. The London series showed a significant improved survival with native liver in those who were treated with AVT (Table 4, Figure 1); the median age at KPE was 64 days in treated versus 70 days in untreated (*p* = 0.07). Four AVT patients were older than 70 days at KPE; all cleared their jaundice.

## 4. Discussion

By collecting and comparing multicentre data from four tertiary centres we showed that the rate of CMV positivity in BA patients varies between centres, but that it is significantly higher than expected for age [1]. Overall, the numbers of CMV-positive BA infants from these European centres were somewhat lower than reported from other parts of the world, and we also noted an ethnic disparity in patients from the largest contributing centre [2,3,4,5,6].

The role of CMV in the pathogenesis of BA is still debated. It is therefore of interest to note that CMV-positive patients differed from CMV-negative patients with regard to age and levels of some biochemical parameters at KPE. This could suggest that CMV-positive patients constitute a specific subgroup, perhaps with later presentation and more pronounced hepatic and bile duct inflammation.

CMV has also been suggested to impact the outcome in BA patients with regard to clearance of jaundice and native-liver survival [3,8,9], and this was consistent with the data from London (Figure 1). Whether these reported differences in survival are due to effects of the viral infection or to the differences in age at KPE remains to be determined. However, the fact that AVT was associated with improved native-liver survival in the London series (Figure 1) suggests that ongoing CMV infection might affect long-term clinical outcomes.

A European survey on post-KPE practices by paediatric surgeons showed that almost 80% of the respondents routinely test for CMV serology and that half of those use AVT if ongoing infection is found [10]. However, there are few data on the effects of AVT in BA from other parts of the world and from the paediatric gastroenterologists/hepatologists who are often are responsible for the follow-up of these patients.

CMV can infect pre-, peri- and postnatally and for the patients reported here we could not determine the timing of transmission. This would clearly be of interest, since there is accumulating evidence to suggest that patients with BA are cholestatic at birth and that the underlying insult(s) occur in utero [11]. One way of advancing knowledge on this matter is to retrospectively analyse CMV-DNA on stored Guthrie cards, which are used for newborn screening of metabolic diseases. In a small study, CMV-DNA was detected on stored Guthrie cards, collected within three days of birth, in only one of 11 BA patients who had an ongoing CMV infection at the time of KPE [12]. That specific patient could therefore be defined as congenitally infected, whereas timing of infection in the remaining patients remained unclear.

Infections such as CMV could be of importance for the pathogenesis of BA not only through a direct viral hit but also by inducing an immune-mediated proinflammatory state causing cellular damage. In support of this hypothesis, Brindley et al. reported that “56% of BA patients had significant increases in interferon-gamma-producing liver T cells in response to cytomegalovirus (CMV), compared with minimal BA responses to other viruses or the control group CMV response. A positive correlation between BA plasma CMV immunoglobulin M (IgM) and liver T-cell CMV reactivity was identified” [13].

In a recently published meta-analysis including 784 patients from nine studies, BA patients with ongoing CMV infection had significantly poorer outcomes than CMV-negative patients, particularly regarding to clearance of jaundice [9]. Theoretically, this negative impact by CMV could be a result of infection at any time point before KPE and the underlying mechanism could either be due to direct viral infection or secondary immune activation [13]. Furthermore, it does highlight an opportunity to detect CMV infection and possibly treat it with AVT if this were found to be of clinical benefit. The data presented herein from London, although not from a randomized, double-blind trial, indicate that such treatment might be beneficial. However, these data need to be confirmed in a placebo-controlled randomized study. Given the overall low incidence of BA, such a study would require a multicentre effort, possibly involving existing disease specific networks, such as the European Rare disease Network (ERN) for rare liver diseases, of which 3 of the 4 participating centres herein are full members, and the Childhood Liver Disease Research Network (ChiLDReN) in North America [14,15].

This study has obvious limitations, one being the retrospective nature and the other being the difference in testing, treatment and follow-up practices between the participating centres. On the other hand, we did assemble data from centres in four European countries that add new information to the topic and also set the stage for future prospective studies.

We conclude that ongoing CMV infection occurs in a considerable proportion of BA patients in European centres of varying size, and that this group differed from CMV-negative patients with regard to age and certain biochemical parameters at KPE and clinical outcomes. We suggest to undertake prospective, multicentre based studies to examine the effects of CMV infection in BA patients and to perform clinical trials of AVT in a randomized placebo-controlled fashion.

## Figures and Tables

**Figure 1 jcm-11-00945-f001:**
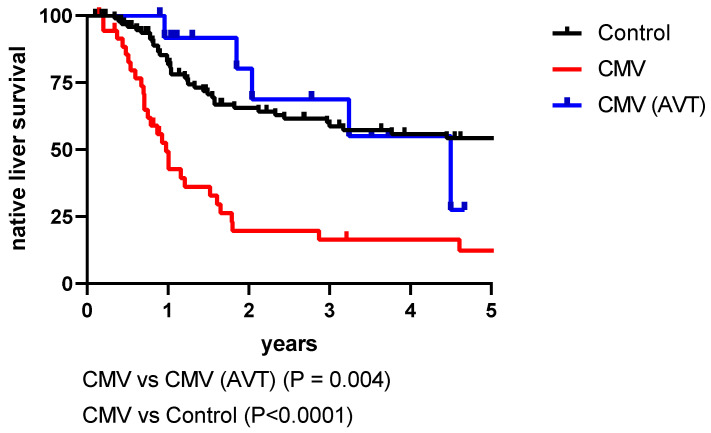
Native-liver survival in cytomegalovirus (CMV) positive patients with biliary atresia, with or without antiviral treatment (AVT) and CMV-negative controls in one of the participating centres (London). Log rank survival.

**Table 1 jcm-11-00945-t001:** Description of the four participating centres with regard to treatment of biliary atresia.

	Budapest	London	Stockholm	Warsaw
**National referral status**	Only centre	Largest centre of three	Largest centre of two	Only major centre
**Time period studied**	2006–2021	2004–2021	2005–2018	1990–2019
**Total national population (million)**	10	65	10	38
**Number of new BA patients/year**	5	25	3	20
**Rate of CMV testing in BA**	67%	75%	100%	76%

CMV—cytomegalovirus; BA—biliary atresia.

**Table 2 jcm-11-00945-t002:** Rate of CMV positivity in tested patients in relation to age at Kasai portoenterostomy (KPE).

Rate CMV Positivity = Positive CMV/Total BA Cases (%) *	KPE < 30 days	KPE 31–70 days	KPE > 70 days
Budapest	0/7	0/22	9/23 (39%)
London	0/7	28/99 (28%)	22/43 (52%)
Stockholm	0	4/16 (25%)	8/21 (38%)
Warsaw	0	21/171 (12%)	39/147 (26%)
Total	0/14	53/308 (17%)	78/234 (33%)

KPE—Kasai portoenterostomy. * Total numbers presented for Budapest, Stockholm and Warsaw. For London patients each CMV positive matched with 2 CMV negative patients for time period.

**Table 3 jcm-11-00945-t003:** Comparison between CMV-positive and CMV-negative BA patients at time of Kasai portoenterostomy.

	Budapest	London	Stockholm	Warsaw
**Age at KPE CMV pos/neg (days)**	88/63 *	69/49 *	78/71 *^p^* ^= 0.10^	79/71 *
**Ethnic disparity**	No	Yes	No	No
**Other anomalies CMV pos/CMV neg**	0%/23% *^p^* ^= 0.18^	16%/18%	17%/12%	n/a
**ALT (IU/L)**	184/122	n/a	114/100	158/145
**AST (IU/L)**	295/173 *	n/a	221/175	n/a
**Bilirubin total (micromoles/L)**	171/144	169/141 *	152/154	168/160
**Bilirubin conj (micromoles/L)**	118/95	n/a	137/125	131/126
**Gamma-GT(IU/L)**	538/443	510/511	279/306	968/768 *
**APRI**	2.7/0.9 *	1.07/0.69 *	1.1/0.8	n/a

ALT—alanine aminotransferase; AST—aspartate aminotransferase; GT—glutamyl transpeptidase; APRI—AST to platelet ratio index; n/a—not available; * Difference statistically significant, *p* ≤ 0.05. All biochemical values are medians.

**Table 4 jcm-11-00945-t004:** Rate and treatment of ongoing cytomegalovirus (CMV) infection in biliary atresia patients at the time of Kasai portoenterostomy.

	Budapest	London	Stockholm	Warsaw
**Rate ongoing CMV at KPE, % (CMV positive/Total BA patient number)**	17% (9/52)	10% ^#^	32% (12/37)	19% (60/321)
**Rate antiviral treatment in CMV pos, %**	44	25	92	>50
**Clearance of jaundice CMV neg/CMV pos (%)**	n/a	71/37 *	44/26	n/a
**Survival native liver CMV neg/CMV pos untreated/CMV pos treated (%)**	28/0/0	60/20/60	40/n/a/26	30/30/30

^#^ estimate (ref. [3]); * Difference statistically significant, *p* ≤ 0.05; n/a not available.

## Data Availability

Data available on request due to restrictions eg privacy or ethical. The data presented in this study are available on request from the corresponding author. The data are not publicly available for ethical and legal reasons.

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
