# Peer review of "Incidence, Impact and Treatment of Ongoing CMV Infection in Patients with Biliary Atresia in Four European Centres"

_jcm, 2022, doi:10.3390/jcm11040945_

Round 1

Reviewer 1 Report

This is a multi-center retrospective study on patients with biliary atresia associated with CMV infection to delineate the impact of CMV infection on clinical conditions and their outcome. They found that the age at KPE was significantly higher and their liver function data tended to be worse in CMV positive pts than in negative pts and the outcome of the CMV positive pts tended to be worse. This paper would add another piece of valuable information to the pediatric hepatologists all over the world who are tackling with this difficult disease. I have two comments.

  1. In an attempt to clarify the timing of CMV infection, the CMV positivity should have been compared in cohorts of each age distribution, eg, before 30 days, 31-70 days and over 70 days after birth as far as the number of pts diminished after division permit the statistical analysis.
  2. It was unfortunate that the role of CMV in the pathogenesis of BA was not fully discussed in this manuscript. I hereby would like to discuss this important issue that I figured out from this data. CMV infection can occur anytime, anywhere after birth and therefore it seems to be natural that the CMV infection is more likely to occur in those who are brought up later to the medical attention for KPE. Conversely, if CMV infection occurred prenatally, involving certain pathogenetic mechanism, those CMV positive pts might have been detected earlier because of worse liver functions.

Author Response

Reviewer 1

This is a multi-center retrospective study on patients with biliary atresia associated with CMV infection to delineate the impact of CMV infection on clinical conditions and their outcome. They found that the age at KPE was significantly higher and their liver function data tended to be worse in CMV positive pts than in negative pts and the outcome of the CMV positive pts tended to be worse. This paper would add another piece of valuable information to the pediatric hepatologists all over the world who are tackling with this difficult disease. We thank the reviewer for this positive remark.

I have two comments.

  1. In an attempt to clarify the timing of CMV infection, the CMV positivity should have been compared in cohorts of each age distribution, eg, before 30 days, 31-70 days and over 70 days after birth as far as the number of pts diminished after division permit the statistical analysis. We thank the reviewer for this suggestion and have added these data in a separate table.
  2. It was unfortunate that the role of CMV in the pathogenesis of BA was not fully discussed in this manuscript. I hereby would like to discuss this important issue that I figured out from this data. CMV infection can occur anytime, anywhere after birth and therefore it seems to be natural that the CMV infection is more likely to occur in those who are brought up later to the medical attention for KPE. Conversely, if CMV infection occurred prenatally, involving certain pathogenetic mechanism, those CMV positive pts might have been detected earlier because of worse liver functions. We agree that the role of CMV in the pathogenesis is of interest to further discuss here and have added a paragraph on this in Discussion.

Reviewer 2 Report

The authors should be applauded for collecting a large number of biliary atresia cases with the aim of studying the interesting and controversial role of CMV infection in the clinical characteristics and outcomes of BA patients.

There is however room for much improvement. The reported data on which the authors base their analyses and significant findings are incomplete. Additional information must be provided to help the reader assess potential causalities, the relevance and strengths of the study findings.

  1. The total number of cases is nowhere to be found in the paper. "A relatively large number of European patients" in not helpful. Data in table 1 (numbers of new patients per year x time periods), suggest that the total number is close to 1000. If so, the numbers in tables 2 & 3 are relatively small.
  2. The different analyses were based on what proportions of what numbers of patients? How much missing data?
  3. Were there systematic differences between tested/non-tested and  treated/non-treated? This is briefly commented in text only.
  4. How many were classified as CMV positive by a single positive CMV-IgM test only? How reliable/significant is this test?
  5. CMV positivity was significantly more common among non-caucasian patients (in London). Was ethnicity registered for all patients? Only in London? For all patients in London?
  6. Why did you chose to display a significantly better outcome for patients receiving AVT, if this was only found in one centre? If this was not true for the whole cohort, are you concerned that your selective presentation might be misleading?

The list of study limitations could thus be expanded. Many of the questions/issues above could potentially be resolved by including a "classic" Table 1 that describe the basic characteristics of the whole cohort and the proportions of patients that are included in different analyses.

Please help the reader. Provide more information. And think twice before reporting significant findings that suggest causality if these are based on only selected subsets of cases.

Author Response

Reviewer 2

The authors should be applauded for collecting a large number of biliary atresia cases with the aim of studying the interesting and controversial role of CMV infection in the clinical characteristics and outcomes of BA patients.We thank the reviewer for this encouraging remark.

There is however room for much improvement. The reported data on which the authors base their analyses and significant findings are incomplete. Additional information must be provided to help the reader assess potential causalities, the relevance and strengths of the study findings. We agree that our data are to a certain extent incomplete. In the revised version we have added some more information and also pointed out the limitations more clearly.

  1. The total number of cases is nowhere to be found in the paper. "A relatively large number of European patients" in not helpful. Data in table 1 (numbers of new patients per year x time periods), suggest that the total number is close to 1000. If so, the numbers in tables 2 & 3 are relatively small. We agree that this was a shortcoming in the previous version. A new table with data on rate of CMV positivity in relation to age at KPE has been added. This allowed us to present more exact numbers.
  2. The different analyses were based on what proportions of what numbers of patients? How much missing data? Please see reply to question #1.
  3. Were there systematic differences between tested/non-tested and  treated/non-treated? This is briefly commented in text only. At least in one of the centres (London) there was an “era effect” in the sense that willingness to test for CMV increased over time. In the same centre, decision whether or not to give antiviral therapy primarily differed with regard to the individual physician in charge. Other than that there were no obvious systematic differences as asked for.
  4. How many were classified as CMV positive by a single positive CMV-IgM test only? How reliable/significant is this test? As stated in Methods “Ongoing CMV infection (CMV positivity) at the time of KPE was defined by any of the following: CMV-IgM positivity in serum, CMV-DNA positivity in urine by PCR, CMV-DNA positivity in serum/plasma/whole blood by PCR.” CMV-IgM positivity, as opposed to CMV-IgG positivity would be reliable to inform that the infant has been infected at some time point before the test is performed. However, its sensitivity may be slightly lower than for example for CMV-DNA positivity in urine which was therefore also included. To ascertain ongoing viral replication, CMV-DNA in serum/plasma or whole blood needs to be tested.
  5. CMV positivity was significantly more common among non-caucasian patients (in London). Was ethnicity registered for all patients? Only in London? For all patients in London? Data on ethnicity were available from all centres, but the mentioned difference was only seen in London, where it was registered for all patients. It should be noted that basically all patients in Budapest and Warsaw were of Hungarian and Polish origin, respectively. Thus, ethnic disparity would have been very hard to prove or rule out in those two centres.
  6. Why did you chose to display a significantly better outcome for patients receiving AVT, if this was only found in one centre? If this was not true for the whole cohort, are you concerned that your selective presentation might be misleading? We agree that this is a limitation and it has been pointed out in Discussion.The data on AVT treatment from London was chosen since this was the only centre with both treated and untreated CMV infected BA patients.

The list of study limitations could thus be expanded. Many of the questions/issues above could potentially be resolved by including a "classic" Table 1 that describe the basic characteristics of the whole cohort and the proportions of patients that are included in different analyses. We agree and have revised the text with regard to study limitations and tried to further clarify the patient numbers.

Please help the reader. Provide more information. And think twice before reporting significant findings that suggest causality if these are based on only selected subsets of cases. We have provided more information as asked for and further pointed out the limitations with regard to analysis of subgroups of patients.

Round 2

Reviewer 2 Report

Thank you for clarifying almost all of my questions. The paper is much improved.